# A mathematical model for the dependence of keratin aggregate formation on the quantity of mutant keratin expressed in EGFP-K14 R125P keratinocytes

**Marcos Gouveia**[1]☯*, **Tjaša Sorčan**[2]☯, **Špela Zemljič-Jokhadar**[3], **Rui D. M. Travasso**[1]‡*, **Mirjana Liović**[4]‡*

**1** Department of Physics, CFisUC, Center of Physics of the University of Coimbra, University of Coimbra, Coimbra, Portugal, **2** AdriaBio Llc, Postojna, Slovenia, **3** Faculty of Medicine, Institute for Biophysics, University of Ljubljana, Ljubljana, Slovenia, **4** Faculty of Medicine, Medical Center for Molecular Biology, Institute for Biochemistry and Molecular Genetics, University of Ljubljana, Ljubljana, Slovenia

☯ These authors contributed equally to this work.
‡ RDMT and ML also contributed equally to this work.
* mgo.fis@protonmail.com (MG); ruit@uc.pt (RDMT); mirjana.liovic@mf.uni-lj.si (ML)

**Data Availability Statement:** All relevant data are within the paper and its Supporting information files.

## Abstract

We examined keratin aggregate formation and the possible mechanisms involved. With this aim, we observed the effect that different ratios between mutant and wild-type keratins expressed in cultured keratinocytes may have on aggregate formation *in vitro*, as well as how keratin aggregate formation affects the mechanical properties of cells at the cell cortex. To this end we prepared clones with expression rates as close as possible to 25%, 50% and 100% of the EGFP-K14 proteins (either WT or R125P and V270M mutants). Our results showed that only in the case of the 25% EGFP-K14 R125P mutant significant differences could be seen. Namely, we observed in this case the largest accumulation of keratin aggregates and a significant reduction in cell stiffness. To gain insight into the possible mechanisms behind this observation, we extended our previous mathematical model of keratin dynamics by implementing a more complex reaction network that considers the coexistence of wild-type and mutant keratins in the cell. The new model, consisting of a set of coupled, non-linear, ordinary differential equations, allowed us to draw conclusions regarding the relative amounts of intermediate filaments and aggregates in cells, and suggested that aggregate formation by asymmetric binding between wild-type and mutant keratins could explain the data obtained on cells grown in culture.

## Introduction

The intermediate filament (IF) cytoskeleton of epithelial cells consists of differentiation-specific sets of type I and type II keratin proteins, which establish a functional filament network that is crucial for the mechanical integrity of cells and tissues [1]. Over the last three decades, hundreds of keratin gene mutations have been discovered and linked to a large group of human diseases manifesting cell and tissue fragility [2].

**Funding:** This work resulted from a collaboration initiated by the COST Action CA15214 (EuroCellNet). M.G. thanks the support of national funds from FCT – Fundação para a Ciência e a Tecnologia, I.P. through grant SFRH/BD/136046/2018. M.G. and R.D.M.T. thank the support of national funds from FCT – Fundação para a Ciência e a Tecnologia, I.P. through projects UIDB/04564/2020 and UIDP/04564/2020. (https://www.fct.pt/). The work was also supported by the CELSA Alliance and ARRS J3-3078 grants to M.L., the Slovenian Research Agency Program Grant P1-0390, and the Republic of Slovenia Ministry of Education, Science and Sport and the EraCoSysMed (JTC-2 2017) "4D-HEALING" grant" (https://celsalliance.eu/, https://www.arrs.si/ and https://www.gov.si/en/state-authorities/ministries/ministry-of-education-science-and-sport/). The funding agents did not play any role in the study design, data collection and analysis, decision to publish, or preparation of the manuscript.

**Competing interests:** The authors have declared that no competing interests exist.

K14p.Arg125, a residue located in the helix initiation motif of keratin 14 (K14), is the most frequently mutated residue in this protein. It is linked to over 30% of reported cases of severe epidermolysis bullosa simplex (EBS), formerly known as severe generalized EBS or EBS Dowling-Meara [3]. Several amino acid changes of this residue have been found in patient-derived cells: R125H, R125C, R125S, R125L and R125P. *In vitro*, they all cause some degree of keratin filament network perturbation, which also manifests in keratin aggregates formation [4–7]. Herrmann and colleagues analyzed the impact of the K14 R125H mutation on *in vitro* keratin filament polymerization and found that under certain conditions, K14 R125H monomers were still able to form functional keratin IFs when mixed with wild-type (WT) K5 at 1:1 ratio [8]. Thus, even if all K14 protein used were mutant, when mixed with normal K5 monomers, it would form rather normal looking keratin filaments. The authors concluded that keratin aggregation, which is often seen both in primary and immortalized mutant keratinocytes, might be the consequence of other mechanistic reasons besides disrupted filament assembly. Nevertheless, the mechanism of keratin aggregate formation is still elusive.

Another interesting aspect of keratin aggregates is that they are not visible in all severe EBS-patient derived keratinocytes and even less so in mild keratin mutants, but they do appear after exposure to some type of external stress. In particular, keratin aggregation is triggered by thermal or hypo-osmotic stress *in vitro* [5, 9]. Others have shown that mutant keratin aggregation may also be reduced if mutant keratinocytes are pre-treated with a chemical chaperone like trimethylamine N-oxide (TMAO) [10, 11]. TMAO and similar compounds act by stabilizing proteins in their native state, protecting them against thermal denaturation and aggregation. The presence of keratin aggregates has also been linked to increased MAPK activity, which place EBS patient-derived cells in a state of constant stress [9].

Previously, we have extended an existing mathematical model for keratin turnover in wild-type cells [12] to account for the emergence of keratin particles in mutant keratinocytes [13]. Our model includes the turnover between soluble, particulate and filamentous keratin forms. We assumed that keratin mutations cause a slowdown in the assembly of an intermediate keratin phase into filaments and *in silico* we have demonstrated this to be enough to cause the loss of keratin filaments and the appearance of keratin aggregates [13].

In the present study, we looked further into the topic of keratin aggregates formation and to the possible mechanisms involved. We addressed this by observing the effect that different ratios between mutant and WT keratin proteins expressed in cultured keratinocytes may have on aggregate formation *in vitro*, and how keratin aggregate formation affects the mechanical properties of cells at the cell cortex [14]. This is an important question, as although the vast majority of EBS cases are inherited in a dominant negative fashion, the actual ratio between WT and mutant keratin protein expressed in cells is not known, and this in turn may also affect the patient's phenotype. Previous AFM and optical stretcher measurements of WT and keratin-free keratinocytes indicated that loss of keratin significantly decreased the stiffness of keratinocytes [15, 16]. However, the re-expression of substoichiometric amounts of K5/K14 normalized cell stiffness and rescued intercellular adhesion defects [15, 17]. When the same mutation we looked at in our study was introduced into mouse cells lacking the entire type I keratin cluster (K14 R125P in human is analogous to K14 R131P in mouse), the mutation had a big impact on cells by affecting IF network formation and by causing keratin aggregation [18]. In addition, this mutation in mouse cells had a larger impact on cell stiffness (causing a decrease) than the actual absence of all type I keratins.

In this study we used stably transfected keratinocyte cell lines we have previously engineered [19], by introducing EGFP labeled K14 constructs containing either the WT cDNA sequence, the R125P mutation, or the V270M mutation (the latter is situated in the L12 linker region of K14 and is linked to the mild, localized type of EBS [20]) into the background of a

WT cell line (NEB1 cells). Here we analysed further the clones with expression rates as close as possible to 25%, 50% and 100% of the EGFP-K14 proteins (WT or mutant). At the same time, we took on a systems biology approach and proposed a reaction network that considers the coexistence of WT and mutant keratins in the cell. With this model we drew conclusions regarding the relative amounts of intermediate filaments and aggregates in the cell. The model consists of a set of coupled, non-linear, ordinary differential equations (ODEs) for which we found the equilibrium state. The model implemented in this work expands the work that was developed over the last two decades [12, 21, 22] on the keratin cycle of normal keratinocytes. Though in Gouveia et al (2020) [13] we modeled the R125P mutation, the interplay between WT and mutant keratins will be addressed for the first time in the present work [5].

## Materials and methods

### Cell lines and culture conditions

We previously engineered several isogenic cell lines by introducing and stably expressing an extra copy of EGFP labeled K14 WT, EGFP labeled K14 R125P and EGFP labeled K14 V270M constructs [19]. In brief, the human K14 wild-type cDNA (NM_000526) was originally cloned into the EGFP-C1 vector (Clontech, USA) and tested by transient transfection of control keratinocyte cells (NEB1). This vector was then used to prepare the mutant construct by introducing the K14 R125P mutation using the QuikChange Site-Directed Mutagenesis Kit (Stratagene, USA). To generate the stable cell lines used in this study, the K14 wild-type and mutant cDNAs were cut out of the EGFP-C1 constructs and re-cloned into the pLEGFP-C1 (Clontech, USA) retro-viral vector using the HindIII and BamHI restriction sites, and the resulting retro-viral wild-type and mutant constructs were used to transfect the NEB1 (control) cell line. After single cell cloning and antibiotic selection with G418, clones were tested by Western blotting for the expression level of the EGFP construct, and the ones that were expressing the EGFP K14 construct (WT and mutant) as close to 25%, 50% and 100% ratio with the endogenous K14 WT (tested by Western blot), were further expanded and used. As none of the 60 analyzed clones had the exact desired ratio, in the end we compared clones with an 18%, 45% and 99% content of EGFP K14 protein (WT or mutant). For easier reference, the clones will still be labeled and referred to in the paper as "25", "50" and "100".

Cells were cultured in RM+ medium: 75% DMEM (Dulbecco's modified Eagle's medium) plus 25% Ham's F12 medium, with added 10% fetal calf serum and additional growth factors hydrocortisone (0.4 mgmL$^{-1}$), cholera toxin ($10^{-10}$ M), transferrin (5.0 mgmL$^{-1}$), liothyronine ($2 \times 10^{-11}$ M), adenine ($1.9 \times 10^{-4}$ M) and insulin (5 mgmL$^{-1}$). All cell lines were fibroblast feeder cell independent and were cultured at 37˚C and 5% $CO_2$. Twenty-four hours before measurements were performed with optical tweezers, cells were seeded in custom-made sample chambers. They were assembled from an acetone cleaned, uncoated, glass cover-slip to which a PDMS insert was attached using plasma.

In proteasome inhibition experiments, MG132 was dissolved in culture buffer at 1 $\mu$M concentration and cells were incubated for 2 hrs at 37˚C and 5% $CO_2$, after which images were taken on a Nikon TE2000 Eclipse inverted microscope.

### Standard protein extraction and Western blotting

All growth media was removed from the culturing flasks and ice-cold RIPA buffer (Sigma-Aldrich, USA) complemented with protease inhibitors was added. Cells were carefully scraped from the bottom with a cell scraper. The cell suspension was collected and centrifuged for 10 min at 12500 rpm and at 4˚C. The protein concentration was determined by Pierce™ BCA Protein Assay Kit (Thermo Fischer Scientific, USA). Equal amounts of protein (20 µg) were run

on a 4–20% gradient polyacrylamide gel. Subsequently, proteins were transferred to a PVDF membrane. Membranes were blocked in 5% (w/v) skimmed milk in TBS-T (20 $mu$M Tris, 150 mM NaCl, 0.02% Tween-20, pH 7.5), which was followed by overnight incubation in mouse primary antibody against keratin 14 (LL001, gift of prof. dr. E.B. Lane) at room temperature. After washing, membranes were incubated with an anti-mouse secondary antibody conjugated to horseradish peroxidase, for 1 hour at room temperature. Proteins were detected with a LAS-400 analyzer after 5 min incubation with the SuperSignal West Pico Plus Chemiluminescent Substrate (Thermo Fisher). Protein quantification was done using ImageJ software.

## Preparation of soluble and insoluble protein fractions

Keratinocytes were grown in culture at 37˚C, 5% $CO_2$, using RM+ medium with 10% FBS (Sigma Aldrich, USA) in T25 flask. The medium was changed every other day till they reached 90% confluence. Cells were then washed twice with 2 ml DPBS (Gibco Thermo Fisher Scientific, USA,) and lysed in 120 $\mu$L RIPA extraction buffer (Thermo Fisher Scientific, USA) supplemented with protease inhibitors (Merck, Sigma Aldrich, USA), which were freshly added prior to use. Cells were carefully scraped from the bottom with a cell scraper and the collected extract was transferred to an Eppendorf tube (Eppendorf, Germany). This process was repeated once using another 80 $\mu$L of RIPA extraction buffer. Total cell lysates were then centrifuged at 12500 rpm and 4˚C for 10 minutes. After centrifugation, the supernatant was transferred to new Eppendorf tube and marked as supernatant (S). The RIPA buffer extraction should result with the majority of keratins being solubilized and thus found in the supernatant. The pellet was then treated as cell extract and re-suspended in high salt buffer B (10 $\mu$L Tris-HCl, pH 7.6, 140 mM NaCl, 1.5 M KCl, 5 mM EDTA, 5 mM EGTA, 1% Triton X-100, with protease inhibitors, prepared as described in Werner et al. [23], homogenized with a pestle (Sigma Aldrich, USA), and centrifuged at 12500 rpm at 4˚C for 10 minutes. The resulting supernatant, which should contain leftover keratin that was not solubized by RIPA buffer (e.g. aggregates. oligomeric particles of intermediate and large sizes), was transferred to a new Eppendorf tube and marked as pellet (P), the RIPA insoluble fraction. All protein extracts were stored at -80˚C. This protocol is a modified version of previously published protocols. [23, 24].

## Real time microscopy

Cover-slips with live cells were assembled into a perfusion open-closed chamber, a miniature climate box system suitable for cultivation and live cell imaging of eukaryotic cells (POC chamber, H. Saur, Germany). The chamber temperature was kept at 37˚C during imaging by a heater. Images (512 x 512 pixels) were collected using a 100x 1.4 NA oil immersion objective on a Nikon inverted microscope TE2000 Eclipse equipped with a motorized Z stage (Applied Precision, USA) and linked to a Micromax CCD camera (Roper Scientific, USA), part of the DeltaVision imaging system (Applied Precision, USA). Time points were collected each 10 seconds during a period of 30 minutes. Images (S1 Raw images) were subsequently deconvolved and analyzed using softWorX software (Applied Precision, USA).

## Optical tweezers and indentation experiments

Cell deformation experiments were performed on an Eclipse Ti inverted microscope (Nikon) equipped with laser tweezers (Tweez 250si, Aresis, Slovenia). The optical tweezers (OT) were set on constant optimal power (laser wavelength was 1064 nm). The laser beam was focused through a water immersion objective (60x, NA 1.00, Nikon) and the sample as well as the objective were heated to 37˚C. A custom-made sample chamber containing adherent keratinocytes was mounted on the microscope stage and silica beads with diameter 5.06 µm (CS01N, Bangs Labs, Fishers,

USA) were added to it. A bead was trapped by optical tweezers and positioned near a single, non-confluent cell, facing the cell side where the cell membrane was most vertical, i.e., the side without extensive lamellipodia. With the help of the piezo microscope stage (Nano-LPS-200, Mad City Labs, USA) the bead centre was positioned approximately 5 nm above the surface. By moving the stage with a constant velocity of 1 μms⁻¹, the cell was pushed into the bead, and its position was monitored in real-time with a digital camera (Zyla 5.5, Andor, Ireland) at 50 fps and by using a custom-written Matlab program (Mathworks, USA). After the bead was pushed for 0.5 μm from the center of the optical trap, the stage was automatically retracted. The force was calculated from $F(t) = -\kappa \Delta x(t)$, where $\kappa$ is the stiffness of the optical trap. The deformation is calculated from the known stage position and the bead displacement: $\varepsilon(t) = x_{stage} - \Delta x(t)$. In the case of OT, fitting of the Hertz-Sneddon model was unreliable because the point of contact was often hard to determine accurately. The cell stiffness was therefore determined as the slope of the linear part of the force-deformation curve. Measurements were performed on at least 35 cells per each cell line.

### Proposed reaction network

In light of what was found in the experimental work presented below, we propose a new reaction network (see Fig 1) that demonstrates how the coexistence of WT and mutant keratin affects the intermediate filament network.

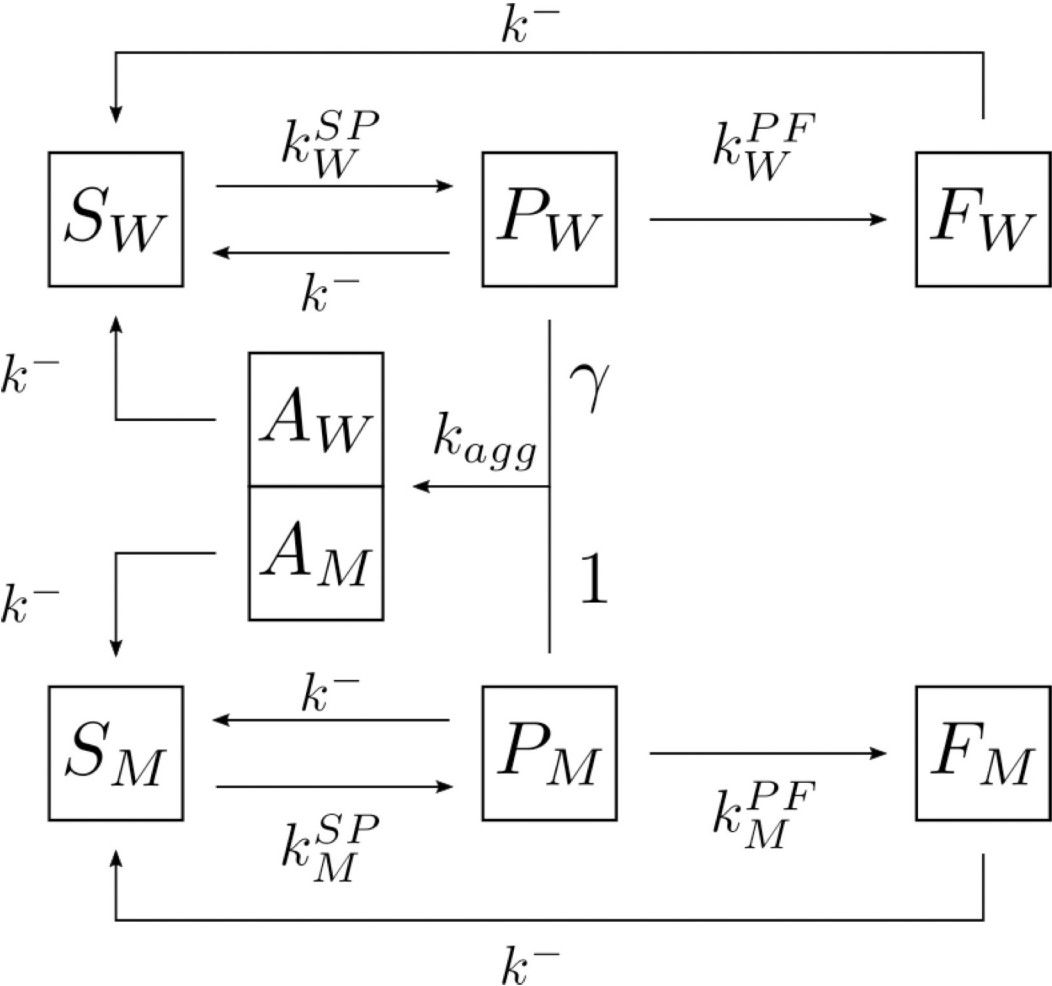

**Fig 1. Graphical diagram of the proposed reaction network.**

Keratins can be found in complex polymeric structures of different sizes. For the keratin polymerization dynamics to be mathematically treatable, and analogously to other keratin models [13, 25], we group the keratin configurations in few different states depending if it forms very small oligomers (soluble keratin), intermediate size oligomers (particulate keratin) or filaments. In the reaction network presented here, we consider 8 different states: 4 for each keratin species (WT and mutant). In the equations describing the reaction dynamics, $S_W$ and $S_M$ will represent the concentration of keratin in the soluble phase (W for wild type and M for mutant), $P_W$ and $P_M$ are the concentrations of keratin in the particulate phase, $F_W$ and $F_M$ the respective concentration in the filamentous phase, and $A_W$ and $A_M$ the respective amount of WT and mutant keratin that is in the form of aggregates. We consider that all depolymerization processes occur at the same rate $k^-$. This assumption is valid if we consider that all the assembled keratin forms are deconstructed one unit at a time. $k_{W/M}^{SP}$ is the rate at which WT or mutant keratin found in the soluble state forms particles, while $k_{W/M}^{PF}$ is the rate at which these same particles assemble to form filaments. We propose that when two particles of different types, one WT and one mutant, assemble with reaction rate $k_{agg}$, they will integrate a keratin aggregate. We use the parameter $\gamma$ to reflect the asymmetric formation of aggregates from WT and mutant particles, which is not necessarily in a 1:1 ratio. A possible asymmetric mechanism of aggregate formation is by binding WT and mutant particles of different typical sizes. These aggregates are not static and they can lose keratin back to the soluble phase.

## Mathematical model

The reaction network presented above in Fig 1 is modeled mathematically by the following system of ordinary differential equations (ODEs):

$$\frac{dS_W}{dt} = -k_W^{SP} S_W + k^-(P_W + F_W + A_W)$$

$$\frac{dS_M}{dt} = -k_M^{SP} S_M + k^-(P_M + F_M + A_M)$$

$$\frac{dP_W}{dt} = k_W^{SP} S_W - k^- P_W - k_W^{PF} P_W - \gamma k_{agg} P_W P_M$$

$$\frac{dP_M}{dt} = k_M^{SP} S_M - k^- P_M - k_M^{PF} P_M - k_{agg} P_W P_M$$

$$\frac{dF_W}{dt} = k_W^{PF} P_W - k^- F_W$$

$$\frac{dF_M}{dt} = k_M^{PF} P_M - k^- F_M$$

$$\frac{dA_W}{dt} = \gamma k_{agg} P_W P_M - k^- A_W$$

$$\frac{dA_M}{dt} = k_{agg} P_W P_M - k^- A_M$$

These equations describe the temporal evolution of the concentrations of the different keratin phases. We assume all reactions can be modeled using first-order kinetics [13, 25] except for the formation of aggregates which depends on the product $P_W P_M$. Since we are only interested in the values of the concentrations when the keratin cycle reaches a dynamical equilibrium, we solve the system of equations until the value of all concentrations are stationary.

To simplify the analysis of the equations, we rewrite the ODEs in a non-dimensional form. We redefine the time variable as $\tau = k^- t$ and replace the concentrations by their respective fractions (indicated in lowercase) of the total amount of keratin in the cell cytoplasm, $K_{total}$,

which is assumed constant. Thus, we are left with the following reformulated system of equations:

$$\frac{ds_W}{dt} = -\lambda_W^{SP} s_W + (p_W + f_W + a_W)$$

$$\frac{ds_M}{dt} = -\lambda_M^{SP} s_M + (p_M + f_M + a_M)$$

$$\frac{dp_W}{dt} = \lambda_W^{SP} s_W - p_W - \lambda_W^{PF} p_W - \gamma \lambda_{agg} p_W p_M$$

$$\frac{dp_M}{dt} = \lambda_M^{SP} s_M - p_M - \lambda_M^{PF} p_M - \lambda_{agg} p_W p_M$$

$$\frac{df_W}{dt} = \lambda_W^{PF} p_W - f_W$$

$$\frac{df_M}{dt} = \lambda_M^{PF} p_M - f_M$$

$$\frac{da_W}{dt} = \gamma \lambda_{agg} p_W p_M - a_W$$

$$\frac{da_M}{dt} = \lambda_{agg} p_W p_M - a_M$$

In these ODEs, the non-dimensional form of the reaction rates are denoted by $\lambda_W^{SP}$, $\lambda_M^{SP}$, $\lambda_W^{PF}$, $\lambda_M^{PF}$, $\lambda_{agg}$, with $\lambda_{W/M}^{SP/PF} = k_{W/M}^{SP/PF}/k^-$ and $\lambda_{agg} = k_{agg} K_{total}/k^-$, where $K_{total}$ is the concentration of all keratin types (WT and mutant) in the cell. Here we set $K_{total} = 1 \times 10^{-3}$ M [12]. The total fraction of mutant keratin in the cell will be denoted by $\chi_M = s_M + a_M + p_M + f_M$.

## Numerical methods

We then proceed to solve these ODEs numerically. In this case, explicit numerical methods such as the Runge-Kutta 4th order method are not adequate due to the stiffness of the equations, since the reaction rates are within a 4 orders of magnitude range. In these situations, the suitable choice is an implicit solver with an adaptive time-step. Here, we chose the LSODA initial-value problem integrator that combines the Adams-Bashforth explicit multistep method with an implicit Backwards Differentiation Formula of the second order [26]. LSODA also includes an adaptive time-step based on the values of the derivatives at each step [27]. The SciPy (Python 3.X.X) implementation of the LSODA algorithm was adopted to solve the equations due to its ease of use and high performance for small systems with a dense Jacobian [28]. The simulations were run until the stationary state. We considered that the stationary state is numerically reached when the absolute value of the time derivatives of all keratin fractions were below $1 \times 10^{-10}$ s$^{-1}$.

## Results and discussion

### Mutant keratin aggregates are dynamic

One characteristic of mutant keratin aggregates is that they are not static: they form, grow and disappear in the cytoplasm of keratin mutant cells [29–32]. Fig 2 presents steps in EGFP labeled keratin filament dynamics showing the formation and disappearance of keratin aggregates in cells expressing a severe mutation (EGFP-K14 R125P construct) at 50% ratio with the endogenous K14 WT protein. The arrows indicate some of the aggregates, which appear at the cell periphery (Fig 2A), grow in size by merging with other aggregates (Fig 2B), and then quickly disappear within minutes, (Fig 2C and 2D) on their inward trajectory towards the

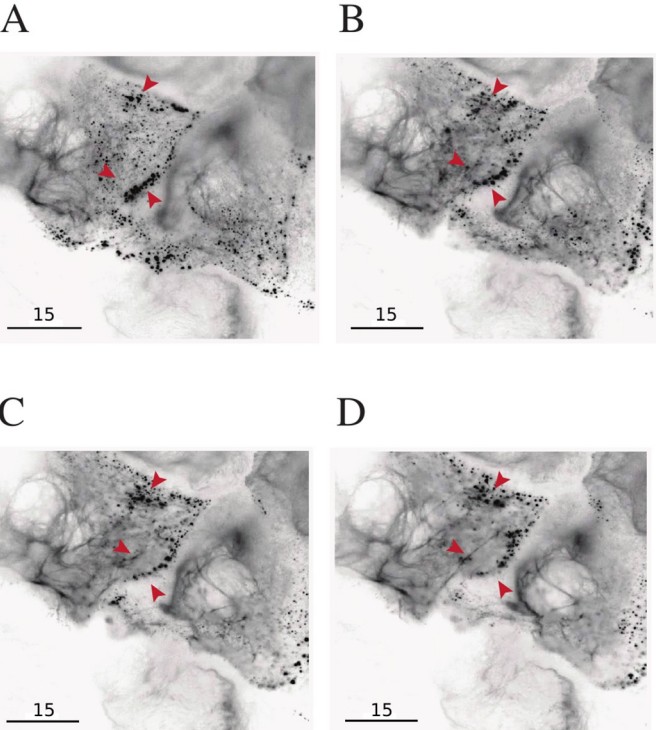

**Fig 2. Fluorescently labeled keratin aggregates are dynamic.** Steps in EGFP labeled keratin filament's dynamics. (A) Aggregates, some marked by red arrows, appear at the cell periphery. (B) Aggregates merge and grow in size (arrows). (C) Larger aggregates disappear within minutes of their inward trajectory to the nucleus (arrows). (D) New filaments can be seen in areas where aggregates were visible previously.

nucleus (see also S1 Raw images). At the same time, where keratin aggregates were previously visible in large numbers, the keratin filament network remodeled and filaments are now visible (Fig 2D).

## Proteasome inhibition induces keratin aggregates accumulation in keratin mutant cells

It has been shown that the activity of the ubiquitine-proteasome system may affect disease severity through keratin aggregate turnover [33]. As in our live cell imaging experiments the EGFP-K14 protein containing keratin aggregates not only accumulated in the cytoplasm but also quickly disappeared, we decided to apply MG132, a proteasome inhibitor, to test whether the disappearance of EGFP labeled keratin aggregates is linked to proteasomal activity. In Fig 3 we show representative images of cells before and 2 hours after treatment. After the inhibitor was administered, aggregates grew in size and accumulated in cells (with about 10% of cells with aggregates before treatment vs. 99% of cells with aggregates after the 2hr treatment with MG132), demonstrating that indeed the EGFP K14 R125P keratin aggregates in our mutant keratin expressing cells are actively disposed of, and therefore this process is both dynamic and reversible.

To look more closely at these aggregates, their possible dependence on the amount of mutant keratin expressed and how their accumulation may affect cell stiffness, we performed clonal selection of our stably transfected EGFP cell lines [19]. Fig 4 shows a Western blot with the ratios between the EGFP labeled K14 construct (WT or mutant) and the endogenous K14

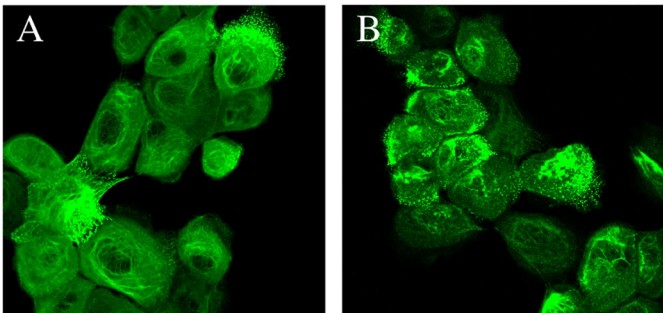

**Fig 3. Proteasome inhibition leads to the accumulation of fluorescently labeled keratin aggregates.** (A) EGFP-K14 R125P cells before treatment with MG132 proteasome inhibitor. Aggregates are visible only within some cells. Aggregates are visible only in about 10% of cells. (B) EGFP-K14 R125P cells after 2 hours of treatment with 1 $\mu$M MG132. Aggregates are present in 99% of cells, indicating that the proteasome system has a leading role in the turnover of these aggregates.

WT protein in clones selected for further analysis. As shown, the selection of clones with closely matching content of K14 was only partially possible, so we decided to select clones that had the desired ratio between the EGFP labeled and endogenous K14 WT, as close as possible to 25%, 50% and 100%. For each type of clone we selected both the two mutants (EGFP-K14 R125P and the EGFP-K14 V270M expressing cells) and the appropriate control cell line expressing EGFP-K14 WT protein (Fig 4).

## Lower cortical stiffness correlates with the presence of keratin aggregates

We hypothesized that the presence of keratin aggregates affects cortical stiffness, and so cell stiffness of clones from Fig 4 was tested with optical tweezers. In order to obtain a reliable estimate of how the percentage of expressed mutant K14 protein may affect cell stiffness, we performed measurements on at least 35 different cells from each cell line. The results are statistically significant and are summarized in Fig 5. Surprisingly, the stiffness of cells where all K14 is derived from the EGFP-K14 R125P mutant construct (100% content, Fig 5A), did not significantly differ from the stiffness of the control cells expressing 100% EGFP-K14 WT, and their means lie within comparable SEM ranges. A similar result was also obtained for cells

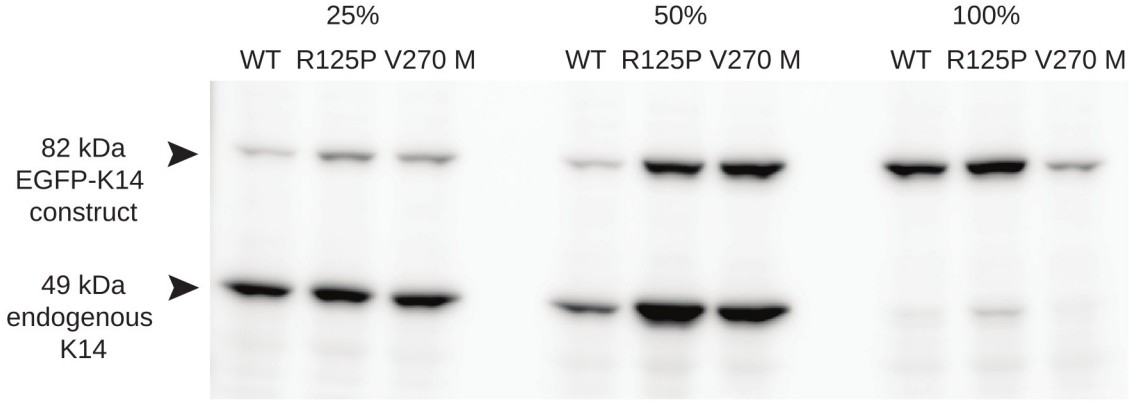

**Fig 4. Western blot analysis of selected EGFP-K14 (WT or mutant) clones.** The upper band belongs to the fluorescently labeled K14 protein, while the lower band to the unlabeled endogenous K14 WT. The numbers 25%, 50% and 100% indicate the ratio between EGFP-K14 to endogenous K14 WT of the clones that were selected for further analysis.

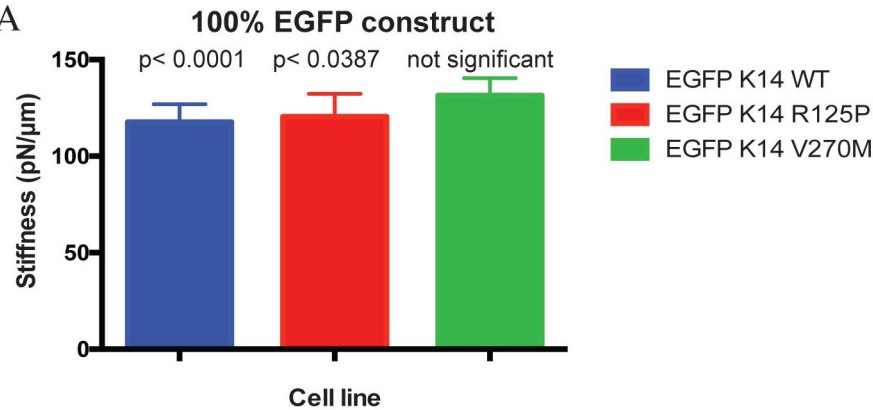

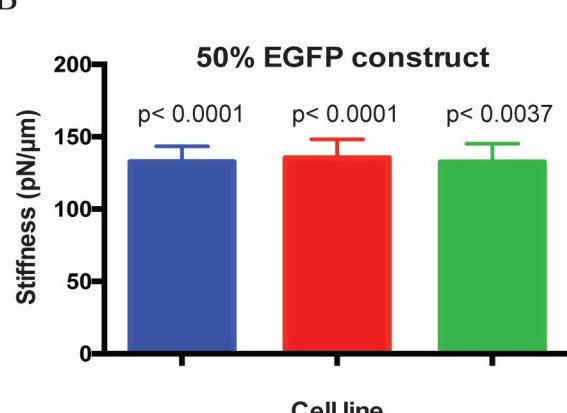

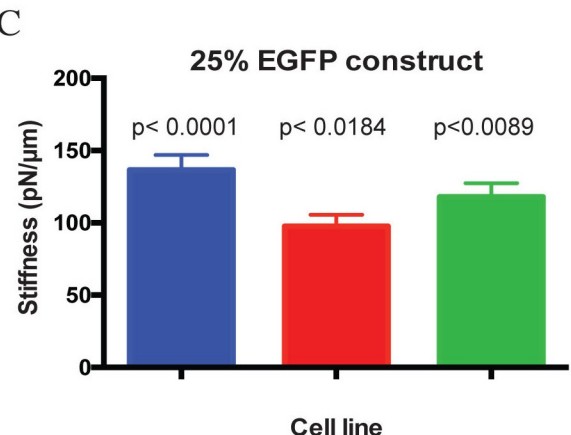

**Fig 5. Correlation between cell stiffness and the expression rate of the EGFP-K14 constructs.** Graphs depict how the expression rates of the EGFP-K14 constructs affect cell stiffness in different EGFP-K14 cell lines. All 100% EGFP-K14 cell lines (A) and 50% EGFP-K14 cell lines (B) do not display any major difference in cell stiffness. In striking contrast to this, there is a significant difference in cell stiffness between the 25% EGFP-K14 cell lines (C). In particular, the 25% EGFP-K14 R125P cells are on average 30% softer than the corresponding control 25% EGFP-K14 WT cells.

A

| Cell line | EGFP K14 R125P 100 % | EGFP K14 R125P 50 % | EGFP K14 R125P 25 % |
|---|---|---|---|
| No. of cells w/out aggregates | 353 | 207 | 288 |
| No. of cells with aggregates | 3 | 25 | 65 |
| Total no. of cells | 356 | 232 | 353 |
| % of cells with aggregates | 0.8 | 10.8 | 18.4 |

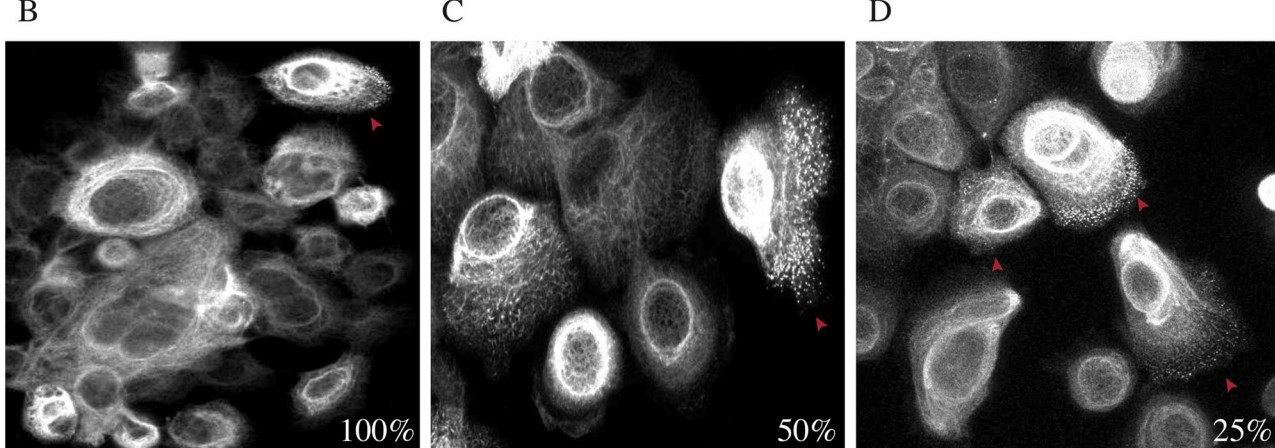

**Fig 6. Quantification of aggregates in cells of EGFP-K14 R125P cell lines.** (A) Summary of data, pointing out to the fact that the least cells with aggregates were seen in the 100% EGFP-K14 R125P cell line (0.8%), followed by the 50% one (10.8%), while the most cells with aggregates were seen in the 25% EGFP-K14 R125P cell line (18%). In the panels below are representative images (fields of view) of each of these cell lines (B-100%, C—50%, D—25%). The arrows point out cells that are representative for cells with aggregates.

expressing 50% of the EGFP-K14 protein (Fig 5B), while a major difference in cell stiffness was detected for the clones where the EGFP-K14 R125P keratin is expressed at a 25% ratio to the endogenous K14 WT (Fig 5C). In this case the EGFP-K14 R125P cells were about 30% less stiff than the EGFP-K14 WT control ones. These results were surprising as a shift in cell stiffness was predominantly expected for cells expressing 100% and 50% EGFP-K14 R125P protein due to the high content of mutant keratin in these cells.

We then also quantified the number of cells with keratin aggregates in these cell lines. The results are summarized in Fig 6. Lower cell stiffness did correlate with the presence of keratin aggregates, however the number of cells with keratin aggregates was inversely proportional to the expression level of EGFP-K14 R125P protein. Namely, aggregates were detected in the cytoplasm of up to 18% of keratinocytes with a 25% ratio of EGFP-K14 R125P, in comparison to 11% of cells with a 50% ratio of EGFP-K14 R125P and only 0.8% of cells expressing EGFP-K14 R125P protein at a 100% ratio. Representative images of the observed differences are shown in Fig 6B (100% EGFP-K14 R125P), C (50% EGFP-K14 R125P) and D (25% EGFP-K14 R125P).

## Endogenous K14 WT is sequestered in the cell pellet

We further analyzed the total cell extract (RIPA extract) and remaining cell pellet (further treated/extracted by high salt buffer) fractions of our different EGFP-K14 cell lines on Western blot (Fig 7) by testing them for K14. This revealed that the pellet fraction of the 25% EGFP-K14 R125P mutant contains a large amount of endogenous K14 WT, and just a trace amount

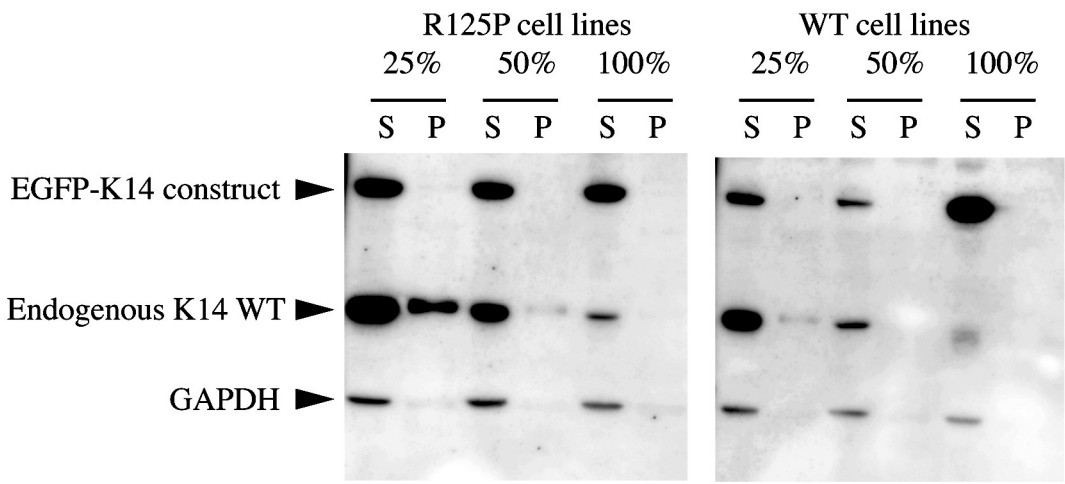

**Fig 7. Western blot of K14 content in EGFP-K14 WT and R125P mutant cell lines in the total cell extract (RIPA supernatant) and the pellet fraction (after extraction with a high salt buffer).** The upper band belongs to the protein derived from the EGFP-K14 construct, while the lower band to the endogenous K14 WT. 25%, 100% and 50% indicate the different EGFP-K14 clones, while S marks the supernatant fraction and P the pellet fraction of the total cell extract(s). Surprisingly, the 25% EGFP-K14 R125P cell line, which also displayed the lowest cortical stiffness and highest number of cells with keratin aggregates, contains a significant amount of endogenous K14 WT in the pellet fraction (over an order of magnitude more than the EGFP-K14 R125P protein in the band above). All other cell lines, including the 50% and 100% EGFP-K14 R125P ones have either no or only trace amounts of both EGFP labelled and endogenous K14 in the pellet fraction.

of the EGFP-K14R125P protein (Fig 7). At the same time, all the EGFP-K14 WT cell lines have little or no endogenous K14 present in the high salt extracts. We therefore reason that unlike the majority of WT and mutant keratin protein, which under RIPA buffer extraction normally ends up in the supernatant of a total cell extract (see also 100% and 50% clones in Fig 7), the large amount of endogenous K14 WT found in the cell pellet in the case of the 25% EGFP K14 R125P cell line may be due to it being sequestered within keratin aggregates, which may be of lower solubility than the filaments.

As these results were unexpected we decided to use a systems biology approach to try to analyze this phenomenon *in silico* with an enhanced mathematical model. This model will not look into the variability in aggregate concentration within a population of cells with similar keratin mutant levels (which will be the focus of a future work), but it will explore possible aggregation mechanisms that allow for the existence of a peak in aggregation at low mutant keratin fractions, as observed in Figs 6 and 7.

## Aggregates formed by asymmetric binding of WT and mutant keratins lead to a maximum in aggregate accumulation at small molar fractions of mutant keratins

The model introduced in this work presents a minimal mathematical description of keratin filament and keratin aggregate formation that considers asymmetric binding between WT and mutant keratins. The values used in the simulation for the reaction rates $\lambda_W^{SP}, \lambda_M^{SP}, \lambda_W^{PF}, \lambda_M^{PF}$, have either been measured experimentally or fitted to data by other mathematical models [12, 13] (see Table 1). According to these reaction rates, the formation of mutant filaments is a slower process than the formation of WT filaments. The reaction rate associated with the formation of keratin aggregates, $\lambda_{agg}$, and the factor representing the asymmetry in binding between WT and mutant particles to form aggregates, $\gamma$, have not been measured experimentally. As such,

**Table 1. Parameters used in the mathematical models (see also S1 Document).**

| Process | Parameter | Value (s$^{-1}$) | Adimensional | Reference |
|---|---|---|---|---|
| Form WT particles | $k_W^{SP}$ | $1.0 \times 10^{-3}$ | $\lambda_W^{SP} = 1$ | [12] |
| Form mutant particles | $k_M^{SP}$ | $1.0 \times 10^{-3}$ | $\lambda_M^{SP} = 1$ | [13] |
| Form WT filaments | $k_W^{PF}$ | $1.0 \times 10^{-1}$ | $\lambda_W^{PF} = 100$ | [12] |
| Form mutant filaments | $k_W^{PF}$ | $5.0 \times 10^{-4}$ | $\lambda_M^{PF} = 0.5$ | [13] |
| Depolymerization | $k^-$ | $1.0 \times 10^{-3}$ | 1 | [12] |

we will explore how the different values for these two parameters affect the ratio between the fraction of keratin found in aggregates and in filaments. This ratio between non-soluble particulate keratin phases and keratin filaments will be denoted by $Q = (a_W + a_M + p_W + p_M)/(f_W + f_M)$.

In Fig 8A we present a typical plot in the stationary state of $Q$ as a function of the total fraction of mutant keratin in the cell, $\chi_M = s_M + a_M + p_M + f_M$. With the parameters in Table 1, we always observe almost no aggregate formation when there is no mutant keratin, since when $\chi_M = 0$ we obtain $Q = p_W/f_W \approx 0$. Similarly, for $\chi_M = 1$, there is no aggregate formation and the fraction of particulate keratin in the cell, $Q = p_M/f_M$, is small since the mutant keratin is still able to form a filament network. For intermediate values of $\chi_M$ we will observe the accumulation of aggregates. For a particular value of mutant keratin in the cell, $\chi_M^{max}$, the observable fraction of aggregates reaches its maximum value, $Q^{max}$.

The values of $\chi_M^{max}$ and $Q^{max}$ obtained in the simulation will depend on both $\gamma$ and $\lambda_{agg}$. In Fig 8B we plot the value of $\chi_M^{max}$, the molar fraction of mutant keratin that leads to the maximum observable fraction of keratin aggregates, as a function of $\gamma$ and $\lambda_{agg}$.

For low $\lambda_{agg}$, the maximum in the observed aggregate fraction is obtained when the mutant keratin is above 50% of the total keratin in the cell, independently on the value of $\gamma$. When the process of aggregate formation is slow, the amount of aggregates in the steady state will directly depend on the size of the pools of particulate WT and mutant keratins in the cell. And so, since the formation of mutant filaments is a slower process than the formation of WT filaments, the fraction of mutant keratin in the particulate phase is higher than the fraction of WT keratin in the particulate phase. Therefore, the most effective way to increase the fraction of keratin in the particulate phase in the cell is to increase the fraction of mutant keratin. Consequently, aggregate formation will peak at keratin mutant fraction above 50%.

On the other hand, if the reaction rate of aggregate formation is high, we obtain the maximum in the observed aggregate fraction for mutant keratin below 50% of the total keratin in the cell, if $\gamma \gtrsim 3$. At high $\lambda_{agg}$, aggregate formation is faster than filament formation. Therefore, if $\gamma$ is large, adding WT keratin is very efficient in increasing the amount of aggregates in the cell, since it readily reacts with mutant keratin in a ratio of $\gamma$ to 1. However, we observe that if $\gamma \lesssim 1$, $\chi_M^{max}$ is always above 50%.

## There is a region of parameter space where the experimental results can be reproduced by the mathematical model

The experimental results of Fig 6 discussed above show that the maximum of keratin aggregates is measured for the 25% ratio EGFP-K14 R125P cell line. We identify in Fig 8B the range of parameters where $\chi_M^{max}$ lays between 15 and 30%. We observe that for values of $\lambda_{agg}$ in the order of $\approx 10^3 - 10^4$ (corresponding to $k_{agg}$ on the order of $10^3 - 10^4 \, M^{-1}s^{-1}$ and for $\gamma > 8$ (including the order of magnitude of the ratio between WT and mutant prevalence in the

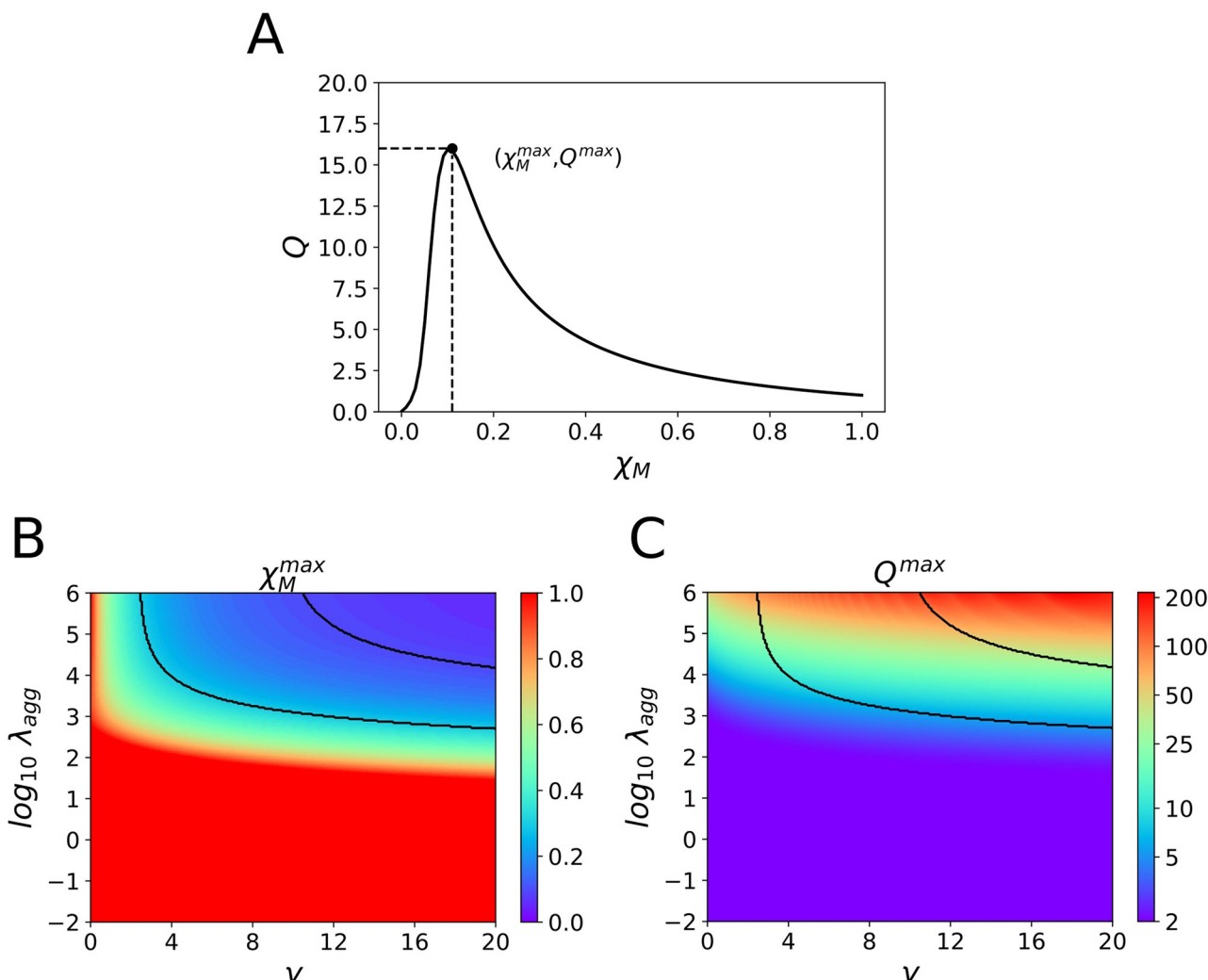

**Fig 8. Fraction of keratin in aggregates in the stationary state obtained by the simulation.** (A) Typical plot of the observable fraction of keratin aggregate in the cell in the stationary state, $Q$, as a function of the total fraction of mutant keratin in the cell, $\chi_M$. (B) Molar fraction of mutant keratin that leads to the maximum observable fraction of keratin aggregates, $\chi_M^{max}$, as a function of $\gamma$ and $\lambda_{agg}$. When the reaction rate to form aggregates is high and if $\gamma \gtrsim 3$ we obtain the maximum in the observed aggregate fraction for mutant keratin below 50% of the total keratin in the cell. The full lines border the region where $0.15 < \chi_M^{max} < 0.30$. (C) Maximum observable aggregate fraction, $Q^{max}$, as a function of $\gamma$ and $\lambda_{agg}$. $Q^{max}$ increases sharply with $\lambda_{agg}$. The full lines border the region where $0.15 < \chi_M^{max} < 0.30$.

pellet observed in the data of Fig 7), the maximum of aggregates is always obtained for a ratio of mutant keratin close to the one that leads to the maximum of aggregates in the experiments on cells. For smaller values of $\gamma$ between 3 and 8, we still observe that a higher value of $\lambda_{agg} \approx 10^4 - 10^6$ (corresponding to $k_{agg}$ on the order of $10^4 - 10^6$ M$^{-1}$s$^{-1}$) leads to a peak in aggregate formation for 15–30% ratio of mutant keratin.

## Computational results predict that for cells with 25% EGFP-K14 R125P the amount of keratin in aggregates is one order of magnitude greater than the keratin filament network

In Fig 8C we plot the value of the maximum observable aggregate function, $Q^{max}$, as a function of $\gamma$ and $\lambda_{agg}$. Expectably, we obtain a larger accumulation of aggregates for larger

$\lambda_{agg}$. Strikingly, we observe that in the region of parameters with $\chi_M^{max}$ on the vicinity of $15 - 30\%$ (bordered by the black lines in Fig 8C), and for $\gamma > 10$, the value $Q^{max}$ is in the $\approx 10 - 50$ range, depending on the value of $\lambda_{agg}$. This means that if we look at the keratin distribution inside a 25% EGFP-K14 R125P cell, the model predicts there will be about one order of magnitude more keratin in aggregates than in filaments (see S1 Document).

## Conclusion

Our study advances the knowledge in the field by analyzing the effect that different quantities of mutant keratin protein expressed have on cell stiffness and the presence of keratin aggregates. We show that the two processes are linked, as the larger the number of cells with aggregates, the lower the stiffness of the cells' cortical area. Our data also indicates a link between the amount of mutant protein expressed and the presence of aggregates.

We also show that the process of keratin aggregation in EGFP-K14 R125P mutant keratin expressing cells is readily dealt with by the proteasome system, as the inhibition of the proteasome with MG132 causes accumulation of fluorescently labeled keratin aggregates. Thus, EGFP labeled keratin aggregates do not accumulate in cells in the same way as in some other human diseases of protein aggregation, like Alzheimer's or Parkinson's disease, where they lead to cell death [34]. Nevertheless, these keratin aggregates appear to be insoluble in denaturing extraction buffers and to be able to "trap" significant amounts of endogenous WT protein.

Another interesting aspect of the K14 R125P mutation is that although this is a severe mutation, in our experiments even an EGFP labeled K14 R125P protein is able to form functional filaments when it is present at concentrations close to 100% of total K14 keratin, which is reminiscent of the results that Herrmann and colleagues [8] obtained on purified keratin protein in *in vitro* polymerization experiments.

The fact that keratin aggregates do not spontaneously appear in all cases of severe keratin gene mutations but that exposure to some kind of physiological stress causing protein misfolding triggers their formation, suggests that the mechanism of aggregate development is linked to basic laws of folding kinetics and native state stability of (keratin) protein. Consequently, different (keratin) mutations may or may not affect protein folding in the same way. In general, mutations create new interaction patterns between amino acids in a protein folding sequence, which destabilizes it and manifests as slowing down of the folding reaction and the formation of transition states, including aggregates [35]. We have taken into account this in our mathematical model of keratin dynamics in cells expressing mutant keratin protein [13], and in this study we have extended this model further by examining different scenarios that are needed for aggregates to form. We then compared the results of mathematical modeling with the data we obtained in experiments on cells. Against our expectations, our *in vitro* experiments on keratinocytes expressing different amounts of EGFP labeled mutant and WT K14 protein showed a maximum of keratin aggregation in cells when the mutant K14 protein is expressed at concentrations around 25%, while the least aggregation is obtained at concentration over 50% of mutant vs. endogenous WT keratin. We would like to stress that keratin aggregates are not stable nor static structures like aggregates seen in some other protein aggregation diseases (e.g. Alzheimer). Instead, they are dynamic and constantly going through turnover, both produced and degraded by the cells at any given point in time (see S1 Raw images, where keratin aggregates slowly disappear and filaments are being formed instead). In addition, we have not performed any selection of cells during our stiffness measurements, but cells were measured at random, so the cell stiffness data we presented on the graphs in Fig 5 are average values of the measurements performed. The peek of aggregate concentration at low mutant keratin levels was reproduced in our mathematical model if we hypothesized that for

keratin aggregates to form, an asymmetric binding of WT and mutant keratin is needed. The extensive exploration of minimal models where WT and mutant keratins bind at a 1:1 ratio predicted, opposite to what was observed experimentally, a maximum of agglomerate formation at mutant keratin fractions above 50% (see S1 Document). We also explored the hypothesis of a nucleating oligomer formed by both WT and mutant keratin that could grow by the addition of WT keratin. This second hypothesis also did not reproduce the experimental results (see S1 Document). Interestingly two previous studies on vimentin, an intermediate filament protein that builds homopolymeric filaments, bear similarities to our mathematical model's explanation of the findings we obtained on the K14 R125P mutation. Namely, about 25% of mutant vimentin is enough to cause disruption of the endogenous filament network [36], while the influence of the same vimentin mutations on cataract formation in the eye lens of mice [37] showed that animals expressing less than 30% of mutant vimentin had in their tissues cytoplasmic vimentin inclusions that also contained endogenous WT vimentin.

Therefore, our mathematical model predicts that aggregates are the result of the asymmetric binding of WT and mutant keratins, with a ratio $\gamma$: 1, where $\gamma > 1$. This agrees with the imbalance between the two types of keratin recovered from the pellets. Finally, when the fraction of mutant keratin is such that the concentration of aggregates is maximum, the model predicts that the concentration of keratin in the aggregates is at least one order of magnitude larger than the amount of keratin in the filaments in the cell. Therefore, in these conditions, the keratin aggregates will be ubiquitous in the cell, just as observed. In a future study we will address quantitatively the variability observed in keratin aggregation at the cell level. Introducing variability in the reaction rates of the model will also permit to better explore quantitatively these measured amounts of keratin.

## Supporting information

**S1 Document. Parameter value estimation and alternative models.**
(PDF)

**S1 Video. Dynamics of keratin aggregates inside keratinocytes.**
(MPG)

**S1 Raw images. The original raw images of the two Western blots from Figs 4 and 7.**
(PDF)

## Acknowledgments

M.G. and R.D.M.T. thank fruitful discussions with Armindo Salvador and Tânia Sousa.

## Author Contributions

**Conceptualization:** Marcos Gouveia, Tjaša Sorčan, Špela Zemljič-Jokhadar, Rui D. M. Travasso, Mirjana Liović.

**Data curation:** Marcos Gouveia, Tjaša Sorčan, Špela Zemljič-Jokhadar.

**Formal analysis:** Marcos Gouveia, Tjaša Sorčan, Špela Zemljič-Jokhadar, Rui D. M. Travasso, Mirjana Liović.

**Funding acquisition:** Rui D. M. Travasso, Mirjana Liović.

**Investigation:** Marcos Gouveia, Tjaša Sorčan, Špela Zemljič-Jokhadar, Rui D. M. Travasso, Mirjana Liović.

**Methodology:** Marcos Gouveia, Tjaša Sorčan, Špela Zemljič-Jokhadar, Rui D. M. Travasso, Mirjana Liović.

**Project administration:** Rui D. M. Travasso, Mirjana Liović.

**Resources:** Marcos Gouveia.

**Software:** Marcos Gouveia.

**Supervision:** Špela Zemljič-Jokhadar, Rui D. M. Travasso, Mirjana Liović.

**Validation:** Marcos Gouveia, Tjaša Sorčan, Špela Zemljič-Jokhadar, Mirjana Liović.

**Visualization:** Marcos Gouveia, Rui D. M. Travasso, Mirjana Liović.

**Writing – original draft:** Marcos Gouveia, Tjaša Sorčan, Špela Zemljič-Jokhadar, Rui D. M. Travasso, Mirjana Liović.

**Writing – review & editing:** Marcos Gouveia, Rui D. M. Travasso, Mirjana Liović.

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
