## [Decision Letter · Decision Letter 0]

17 Aug 2021

PONE-D-21-22878

A Mathematical Model of the Dependence between Keratin Aggregate Formation and the Quantity of Mutant Keratin Expressed in EGFP-K14 R125P Keratinocytes

PLOS ONE

Dear Dr. Travasso,

Thank you for submitting your manuscript to PLOS ONE. After careful consideration, we feel that it has merit but does not fully meet PLOS ONE’s publication criteria as it currently stands. Therefore, we invite you to submit a revised version of the manuscript that addresses the points raised during the review process.

We look forward to receiving your revised manuscript.

Kind regards,

Jian Xu, Ph.D.

Academic Editor

PLOS ONE

Journal Requirements:

Reviewers' comments:

Reviewer's Responses to Questions

**Comments to the Author**

1. Is the manuscript technically sound, and do the data support the conclusions?

Reviewer #1: Yes

Reviewer #2: Yes

2. Has the statistical analysis been performed appropriately and rigorously? 

Reviewer #1: Yes

Reviewer #2: I Don't Know

3. Have the authors made all data underlying the findings in their manuscript fully available?

Reviewer #1: Yes

Reviewer #2: Yes

4. Is the manuscript presented in an intelligible fashion and written in standard English?

Reviewer #1: Yes

Reviewer #2: Yes

5. Review Comments to the Author

Reviewer #1: The paper entitled ``A Mathematical Model of the Dependence between Keratin Aggregate Formation and the Quantity of Mutant Keratin Expressed in EGFP-K14 R125P Keratinocytes" presents a combination of experimental and computational results on the formation of keratin aggregates in keratinocytes.

The paper presents interesting results using a broad collection of methods from cell biology, biomechanics and biomathematics; however, some results are over-interpreted or all data are not presented or properly explained making some rationale hard to follow. The paper needs revisions before publication.

-Total amount of keratin (endogenous + EGFP K14): Have you verified the total amount of keratin in the different conditions (25$\\%$, 50$\\%$ and 100$\\%$)? Does any condition induce an increase/decrease of the total amount of keratin in cells?

- Line 136 (Definition of insoluble fraction (P)): Specify what does P refer to? filaments + particles + aggregates? endogenous only, EGFP K14 only, mix of endogenous and EGFP K14 material?

-Interpretation of $\\gamma$ : $\\gamma$ is used in the model to describe the asymmetric formation of aggregates from wild and mutant particles. From the equations of the model we have: if $\\gamma<1$ (resp. $\\gamma>1$), the per capita rate of formation of $A_W$ from $P_W$ is smaller (resp. larger) than the per capita rate of formation of $A_M$ from $P_M$ for the same concentrations of the other type of particles $P_M$ or $P_W$. The authors write ``$\\gamma$ is used to represent the difference in size between typical WT and mutant particles that are able to bind and form aggregates.''. This sentence is misleading. A simple sentence in the mathematical model section as " $\\gamma$ is used to represent the asymmetric formation of aggregates from wild and mutant particles" should be enough. Related to this issue, problem with the interpretation of results on page 11 and line 427.

- Define $K_{Total}$ in line 206.

- Define the total fraction of mutant keratin in cells $\\chi_M$ in line 212:$\\chi_M=s_M+p_M+A_M+f_M$ with $\\chi_W+\\chi_M=1$

- Results of models are given at stationary state. Specify the criterion used to detect the fact the solutions of the model have reached the steady state value. That is missing in section ``Numerical methods'' starting on line 213.

- Remove the sentence ``Due to the size of the system of equations and to its non-linearity, it is not possible to find the stationary state solution analytically''. By reducing the dimensional system to a 6-dimensional system (using the conservation of law on the total amount), some expression might be obtainable but an only numerical approach as followed in the paper is enough.

- Figure 4: Does the total amount of keratin in cells change when mutants are added?

- Figure 5: At least 35 cells are used to obtain the measurements of cell stiffness. Nothing is mentioned on how these cells are sampled (selected) for each condition. Does the samples used for stiffness measurements are representative of results shown in Figure 6?

- Figure 5: What about the stiffness of cells expressing only endogenous keratin? Does the addition of EGFP K14 construct change the cell stiffness?

- Figure 5: Specify the values referred by stars.

- Figure 5: Change the title of this figure. This title is not representative of the information displayed in the figure.

- Figure 6: Add the results for EGFP K14 WT as a control case.

- Line 283, "..however the quantity of aggregates did not correlate with the expression level of EGFP-K14 R125P protein." How do the authors get this conclusion? No information on the quantity of aggregates is provided, only information about the number of cells with aggregates is available. Rewrite this sentence.

- Figure 7: Add the results for EGFP K14 WT as a control. Usually IF proteins are found mostly assembled in the insoluble fraction. However, for all conditions in presence of mutant, the soluble fraction is larger than the insoluble one. (As previously mentioned, more details should be given on what is the insoluble fraction.)

- In line 307 "..due to it being sequestered within keratin aggregates, which may be of lower solubility than the filaments.". The authors interpret the results from Fig 7 by speculating that aggregates might have a lower solubility than filaments. However, the disassembly rate $k_{-}$ is assumed to be the same for all (particles, aggregates and filaments) in the model.

- Line 319: The definition of Q as the observable fraction of keratin aggregate in the cell cytoplasm is a little misleading. The equation defining Q is the ratio of the proportion of particulate material to that of filamentous material in cells.

- Line 320: Typo in the definition of Q.

- Add a reference to a figure to the sentence starting on line 351.

- On Line 361, give a proper reference to a figure or other instead of just "see above".

- Validation of the model by experiments (paragraph starting on line 349): Figure 6 deals with the number of cells with aggregates, Figure 7 deals with the soluble and insoluble fractions (the insoluble fraction is not clearly defined yet) of endogenous and EGFP K14. But no clear experimental data are shown about the quantity of aggregates in a cell. Add information or explanation or rewrite.

- Further validation of the model by experiments: From Figure 7, information on the total (endogenous + EGFP K14) soluble and insoluble fractions is available. This information is also tractable in the mathematical model. The model should also be able to predict these features too. Knowing the ratio between particulate and filamentous material (defined by Q) is interesting but this information could be accompanied by the distribution of keratin between the soluble and insoluble fractions.

- The experimental results and their interpretation highlight the importance of the solubilization process translated in the model with the term depending on $k_{-}$. A full sensitivity analysis should be carried out to determine the influence of other parameters (not only $\\lambda_{agg}$ and $\\gamma$).

- Line 376: ".. as the more keratin aggregates are present in cells, the lower the stiffness of the cells' cortical area." Similar remark as before, this statement seems to be an over-interpretation of results.

Minor remarks:

- Add a reference to the sentence starting on line 36.

- Typo or missing information on line 82: "(Clontech, , USA)"

- Typo on line 420

- The supplementary material should be polished.

Reviewer #2: The manuscript by Gouveia et. al. entitled “A Mathematical Model of the Dependence between Keratin Aggregate Formation and the Quantity of Mutant Keratin Expressed in EGFP-K14 R125P Keratinocytes” is interesting and well written. The authors investigate how cells with various proportions of mutant keratin and wild type keratin are different. They examine the number of cells exhibiting keratin aggregates, cell stiffness, and the amount of soluble vs insoluble keratin in the cells. Additionally, the authors developed a mathematical model to help explain the surprising experimental results. The paper should be published. I have several comments.

Pg 5 “gamma is used to represent the difference in size …” I think gamma gives the ratio of wt to mutant in the aggregation and stating that would be clearer.

Fig 5 What do the *** mean? It is not explained in the caption

Fig 6 The arrows are not mentioned in the caption

Pg 9 Five times more WT than mutant but only 18 percent of cells show aggregates. You should be able to approximate how much more WT is in the aggregates. Also is the mutant consistent with the 18 percent – the soluble is much darker than the pellet is it greater than 18 percent (since the pellet is more than just the aggregate)?

Fig 7 Why change the order in this figure to 25% 100 % 50%? Also the 100 % had the least aggregate but seems to have more pellet than the 50 %?

Pg 10 Why is Q not (aW+aM)/(fW+fM)? Why include the particulate keratin? It seems figure 8 would make more sense (or at least in the explanation) if the aggregate keratin was reported from the model as well as Q (or instead).

Pg 10 line 372 “magnitude more keratin in aggregates than in filaments” The model says Q should be that not the aggregates (Q is more than just the aggregates).

More explanation of the stiffness results should be included.

Minor issues:

Pg 6 line 203 concentrations is stationary change to concentrations are stationary

Pg 10 Q=(….)/(fM+fM) should be (…)/(fM+fW)

Pg 12 line 420 “model?s”

6. PLOS authors have the option to publish the peer review history of their article (what does this mean?). If published, this will include your full peer review and any attached files.

Reviewer #1: No

Reviewer #2: No

---

## [Decision Letter · Decision Letter 1]

25 Nov 2021

A Mathematical Model for the Dependence of Keratin Aggregate Formation on the Quantity of Mutant Keratin Expressed in EGFP-K14 R125P Keratinocytes

PONE-D-21-22878R1

Dear Dr. Travasso,

We’re pleased to inform you that your manuscript has been judged scientifically suitable for publication and will be formally accepted for publication once it meets all outstanding technical requirements.

Kind regards,

Jian Xu, Ph.D.

Academic Editor

PLOS ONE

Additional Editor Comments (optional):

Reviewers' comments:

Reviewer's Responses to Questions

**Comments to the Author**

1. If the authors have adequately addressed your comments raised in a previous round of review and you feel that this manuscript is now acceptable for publication, you may indicate that here to bypass the “Comments to the Author” section, enter your conflict of interest statement in the “Confidential to Editor” section, and submit your "Accept" recommendation.

Reviewer #1: All comments have been addressed

Reviewer #2: All comments have been addressed

2. Is the manuscript technically sound, and do the data support the conclusions?

Reviewer #1: Yes

Reviewer #2: (No Response)

3. Has the statistical analysis been performed appropriately and rigorously? 

Reviewer #1: Yes

Reviewer #2: (No Response)

4. Have the authors made all data underlying the findings in their manuscript fully available?

Reviewer #1: Yes

Reviewer #2: (No Response)

5. Is the manuscript presented in an intelligible fashion and written in standard English?

Reviewer #1: Yes

Reviewer #2: (No Response)

6. Review Comments to the Author

Reviewer #1: The authors have addressed all my questions and comments and clarified the confusing parts of the original material. I have no further requests.

Reviewer #2: (No Response)

7. PLOS authors have the option to publish the peer review history of their article (what does this mean?). If published, this will include your full peer review and any attached files.

Reviewer #1: No

Reviewer #2: No

---

## [Editor Report · Acceptance letter]

14 Dec 2021

PONE-D-21-22878R1 

A Mathematical Model for the Dependence of Keratin Aggregate Formation on the Quantity of Mutant Keratin Expressed in EGFP-K14 R125P Keratinocytes 

Dear Dr. Travasso:

I'm pleased to inform you that your manuscript has been deemed suitable for publication in PLOS ONE. Congratulations! Your manuscript is now with our production department. 

Kind regards, 

on behalf of

Dr. Jian Xu 

Academic Editor

PLOS ONE